# Associations between Poor Vision, Vision-Related Behaviors and Mathematics Achievement in Chinese Students from the CNAEQ-PEH 2015

**DOI:** 10.3390/ijerph17228561

**Published:** 2020-11-18

**Authors:** Sijia Wang, Xingjie Hao, Xiao Ma, Yong Yu, Lili Wu, Yan Wang, Youfa Li

**Affiliations:** Collaborative Innovation Centre of Assessment Toward Basic Education Quality, Beijing Normal University, Beijing 100875, China; 202031630003@mail.bnu.edu.cn (S.W.); 201831630003@mail.bnu.edu.cn (X.H.); pansymaxiaomix8725_1@hotmail.com (X.M.); yongyu@bnu.edu.cn (Y.Y.); lilywu1016@163.com (L.W.); yukiho.wy@mail.bnu.edu.cn (Y.W.)

**Keywords:** vision health, primary school students, mathematics achievement

## Abstract

*Purpose*: Poor vision is prevalent in school-aged students, especially in East Asia. This can not only cause irreversibly physical health impairments like glaucoma and cataracts, but also the loss of individual education and employment opportunities and deterioration of the quality of life. The present study aims to investigate the associations between poor vision, vision-related risk behaviors, and mathematics achievement in youth from China. *Methods*: The present study included a total of 106,192 Grade 4 students and 70,236 Grade 8 students from the China National Assessment of Educational Quality-Physical Education & Health 2015 (CNAEQ-PEH 2015). We conducted a standard logarithmic visual acuity scale for vision screening, a self-reported questionnaire for vision-related risk behavior and a standardized mathematics assessment for mathematics performance. Poor vision is defined as the visual acuity below 5.0 by using the standard logarithmic visual acuity chart. Linear regression was conducted. *Results*: The prevalence rate of poor vision in China was 37.1% in Grade 4 and 66.2% in Grade 8 in 2015. Students who had poor vision were more likely to have better mathematics achievement than those with normal vision. Reading in bed, insufficient sleep, and screen time during weekdays and weekends were associated with higher odds of poor vision. *Conclusions*: Poor vision was positively associated with mathematics academic achievements, while vision-related risk behaviors such as screen time, homework time and reading in bed were associated with a high prevalence of poor vision in compulsory education cycle students.

## 1. Introduction

Poor vision is recognized as an impairment caused by an interaction of genetic and acquired factors. Poor vision is defined as the visual acuity below 5.0 by using the standard logarithmic visual acuity chart. Poor vision is essentially an impaired sensory system that can quickly develop from slight impairment to a severe one in children [1] and is one of the important causes of preventable blindness in adults. Poor vision can be a prolonged and progressive disability in individuals, can have harmful effects on students’ academic performance, education potential, fundamental movement skills, physical fitness, and mental health [2,3], and results in a substantial burden on society [4]. Recent evidence has demonstrated that school-age students are a particularly vulnerable group since refractive errors and myopia are highly prevalent in school-age students [5,6], especially in Asia. The highest prevalence is found in East China, a study in 2004 showed that the prevalence of myopia is 73.1% in school-age children (aged 5–15) in china [7]. Though inheritance and family characteristics, including a family history of high myopia [8], parental myopia, sex [9], and birth order [10], are crucial influential indicators, most epidemiological studies have demonstrated the significant impact of behavioral factors on individuals’ vision health. The behavioral factor is a modifiable domain to decrease exposure to the prevalent risk factors for poor vision and slow down the injury process leading to blindness.

Educational studies and epidemiological results have documented the high prevalence of poor vision among school-age students because of exposure to prolonged vision-demanding academic tasks and prolonged indoor sedentary lifestyles. Studies based on academic burden documented that students spent most of their school time sedentary with nearly 70% of total school daily time engaged in demanding academic tasks requiring visual input [11]. Students perform reading, writing, sustaining focus, and other relevant tasks in a crowded academic schedule oriented toward achievement tests and national assessments. Students who spend additional time learning, reading and writing are at a higher risk of poor vision than their peers. Surveillance results demonstrated that the average daily time spent on homework and sustained eye focus for 30 min without a break were risk factors for poor vision [12,13]. Some previous studies documented that a higher prevalence of myopia is associated with female sex [6,14]. According to the conclusion of these studies, girls are more likely to engage in indoor close-up reading activities, and lack of outdoor physical activity, which may explain why girls are more prone to poor vision [15].

The relationship between poor vision and academic achievement has long been debated. According to the evidence of some cross-sectional studies and some follow-up studies, a positive association between poor vision and academic performance has been proposed. Rosário found a correlation between mathematics homework and mathematics academic achievement [16]. In a Singapore cohort study, Saw and his colleagues reported that performance in mathematics, English language, native language, and average examination scores were associated with myopia, even after adjusting current reading and IQ test scores, students (aged from 10 to 12-year-olds) who scored better academically were struggling with more severe vision impairments than their peers [17]. The association between poor vision and academic performance may due to the academic-related duration students daily paid. Children who spend more time reading or writing on academic-related tasks may perform better in school tests. In addition, performance in school tests may reflect the total amount of time spent in reading and writing [18].

However, the following latest results from education follow-up studies and epidemiological studies suggested an inconsistent association between academic achievement and vision impairment, students with poor vision may performance even worse than their peers with healthy vision. After controlling several characteristics of the children and their families, data from Young Lives sample showed a significant negative effect of poor visual acuity on mathematics tests, marginally significant for vocabulary by analyzing children aging from 7 to 8 in Peru [19]. Similar results were reported in English learning performance [20]. A prospective cohort study showed that a better visual acuity in seventh grade was significantly associated with higher ninth-grade scores [21], which suggested an interfere with education potential from visual impairment. The adverse association between poor vision and academic performance may be due to the educational inequity caused by poor vision. The majority of vision impairment are due to refractive error (e.g., myopia), which can be effectively addressed with properly-fitted eyeglasses. However, Due to the insufficient attention of students and their parents to visual health, the eyeglasses are taken up at low rates [22,23]. Some researchers claimed that, when uncorrected, the refractive errors can cause symptoms of blurred vision, ghosting, headache, asthenopia, and can also potentially have a negative impact on visual performance while doing academic-related tasks [24]. In fact, the vision impairment of some students is not caused by refractive error, but by amblyopia and other eye diseases. Their best-corrected visual acuity are still below the normal standard, and it is more difficult than other students in preforming academic-related behaviors. They are observed in the disadvantages of changing visual attention rapidly from workbooks to the board, distinguishing colors for important information, and performing movements demanding hand-eye coordination [2,3].

The inconsistent results of the association between poor vision and academic performance are generally accounted for the variation among sample size, nation of samples, age of samples, and also the assessment tools for visual acuity and academic achievement. Based on an integrated view of pathogenic factors and protected factors, visual acuity effects on academic performance is mediated by pathogenic factors, such as duration of close-visual tasks, reading habits, and screen duration, and protectable factors, such as physical activity habits. This hypothesis obtained support from the data analysis procedure and results from the Young Live sample in Peru [19] suggested that the descriptive statistics were not sufficient to investigate the correlations between academic performance and vision health. Path analysis or logistic regression analysis is appropriate to understand the essential correlation between academic performance and vision impairment by including some behavioral habits of students as control variables.

Based on the epidemiological evidence, a healthy lifestyle seems to be an important factor in vision impairment prevention, and these factors include reading and writing habits, academic burden, outdoor physical activity, sleeping, and other vision input-related habits. Reading and writing habits and academic burdens are crucial influential factors for visual health, while outdoor physical activity, sleeping, and vision input-related habits are recognized as protective factors. School-based studies have proposed close-up reading and prolonged screen watching as the key factors that impair students’ vision [25]. Surveillance results from Changsha in China proposed that reading and writing distances of less than 1 foot between the eyes and the reading and writing materials, reading in a moving vehicle, and sustained eye focus for 30 min without a break are risk factors for poor vision [12]. In addition to the learning burden, average daily sleeping time, the average daily time spent on homework, age, and average daily TV time also been reported as positively associated with poor vision in students aged from 9 to 18 from Yunnan Province in China [13]. Similar results of these positive associations between visual-related risk behaviors and visual health have also been observed in school-age students in Beijing [6] and Zhejiang Province [26].

Though some existing studies have investigated the underlying correlations between visual health and vision-related risk behaviors and the correlations between visual health and academic performance separately, an integrated investigation that included visual screening results, vision-related behaviors, and academic performance remains to be conducted. Also, previous large studies on vision health either lacked a national representation or failed to use standardized measures. Also, the high prevalence of poor vision in students from China indicates that Asia students obtain some specific potential factors that increase their vision impairment risk. Previous studies have investigated the socioeconomic and epidemiological effects of vision impairment in Chinese students at the province level, but poor vision prevalence at the national level and the associations between poor vision, vision-related risk behaviors, and academic achievements are sparse.

The China National Assessment of Education Quality (CNAEQ) is the largest continuing and nationally representative education quality assessment in China with authorization from the Ministry of Education of the People’s Republic of China (MOE of PRC) [25]. It targeted Grade 4 students in primary school and Grade 8 students in middle school as the representative populations to ensure a general acquaintance with their learning environment and learning mode, an accurate assessment of reading comprehension, and a reliable measure of written expression. It comprehensively integrates students’ academic performance, daily behaviors, and learning environment, including teacher-related and institution-related behaviors. The academic performance included the essential disciplines of language, mathematics, science, art, physical education and health, and morality. The students’ daily behaviors included the physical activity-related healthy lifestyle, learning habits and experience, time spent on learning-related events, student-teacher relationships, and attitudes from parents. The learning environment included learning opportunities, teaching quality, and support from the school and social environment. The assessment schedule contains a 3-year assessment period with two disciplines each year to continuously provide cross-sectional and longitudinal data of academic achievement and physical fitness, to determine the current implementation of education in primary school and middle school and to periodically explore the facilitators and barriers in the current education system throughout China. There are two disciplines assessed in a particular year, thus, those two assessments share the same sample, and the underlying correlations between can be analyzed. In 2015, China began its first period of the CNAEQ, physical education (PE) and health, titled China National Assessment of Education Quality-Physical Education & Health (CNAEQ-PEH), was assessed at the same time as mathematics, titled China National Assessment of Education Quality-Mathematic (CNAEQ-MA). Further details are described at https://eachina.changyan.cn/or in Wu et al. [27].

The current study employed some assessment results of the CNAEQ-PEH and CNAEQ-MA from their shared sample and explored the associations between poor vision, risk behaviors, and consequences of poor vision on mathematics achievement. The data analysis was based on students’ eye-screening results, standardized mathematics achievement results, and results from the lifestyle-related, self-reported questionnaires of the CNAEQ-PEH 2015 and CNAEQ-MA 2015. It is oriented to provide an overview of poor vision prevalence in Grade 4 and Grade 8 students in China and new evidence of an association between vision health, vision-related risk behaviors, and mathematics achievement. The findings provide important insights into the health and wellbeing of students with poor vision in China and help to investigate the underlying correlations between poor vision, vision-related risk behaviors, and mathematics achievements, enriching the literature on the relationship between visual health and academic achievement in children and adolescents.

## 2. Materials and Methods

### 2.1. Data Source

Data in the present study were obtained from the first assessment cycle of the CNAEQ, including the, administered on 18 June 2015. The overall effective sample size is 111,173 Grade 4 students from 4015 elementary schools and 72,243 Grade 8 students from 2461 middle schools across 323 counties in China. The CNAEQ-PEH consists of a battery of field tests for physical fitness and health and a specific indicator questionnaire oriented for activity-related healthy lifestyles and relevant health-related habits. A vision screening is included in the field test battery to estimate the students’ visual accuracy because of the high prevalence of poor vision and its progression at young ages. Reading habits, screen time, and sleep duration are included in the questionnaires as measures of health-related indicators. The CNAEQ-MA consists of a standardized mathematics achievement assessment and a specific questionnaire for indicators. Homework time is a measure oriented to describe the learning effort but also a sensitive indicator to partly estimate the daily duration of close-visual tasks. This analysis finally enrolled 106,192 Grade 4 students aged 9–11 years old and 70,236 Grade 8 students aged 13–15 years old into data analysis (age, mean ± SD, 10.38 ± −0.54 in Grade 4 students, 14.43 ± 0.59 in Grade 8 students; gender, proportion of boys, 52.8% in Grade 4 students, 53.2% in Grade 8 students).

### 2.2. Measures

#### 2.2.1. Visual Acuity

In the current study, a standard logarithmic visual acuity chart [28] with a lightbox was used to examine students’ visual acuity. Standard logarithmic visual acuity chart was designed by a Chinese researcher, which has been used uniformly in china since 1990. The standard logarithmic visual acuity chart uses a five-point notation in a range from 4.0 to 5.3 [29]. This vision chart was composed of 14 rows of optotypes with 24 mm line spacing. Each optotype was a squared “E” with all paths equal in width and length, rotated 90 degrees in 4 different orientations. The optotypes increased in size with a geometrical progressive rate from the bottom to the top row. In the standard logarithmic visual acuity chart, there’s a record on the right of each optotype, which represents the result of one’s visual acuity. A picture of the standard logarithmic visual acuity chart is shown below (see Figure 1).

Students were tested individually by opticians or trained school doctors following standard logarithmic vision tests to eliminate any interference and to avoid inaccurate inspection results. Students were asked to stand 5 m away from the vision chart, and the row representing the 5.0 score is equal to the student’s eye level. During the vision test, students cover one eye and identify the direction of the open side of “E” with one eye. Students can use language or gestures to indicate the direction of the open side. The tester should use a stick to randomly point to an optotype in the row that represents the score of 5.0. If the number of mistakes reaches 2, the tester will check the upper row. Instead, the tester will check the next row if the number of mistakes is less than two. In the same way, there should be no one mistake in each row representing 4.0–4.5. No more than 2 mistakes can be admitted in each row representing 4.6–5.0, and no more than 3 mistakes in each row representing 5.1–5.3. If the student fails to meet this standard, the test is terminated and the score on the preceding row is recorded as the student’s vision acuity. The worse score from either eyes was used for the analysis in this study. According to the criteria from the National Survey and Research Manual on Students’ Physical Fitness and Health [30] in China, screening results were divided into four levels, including normal (≥5.0), mild poor vision (4.8~4.9), moderate poor vision (4.6~4.7), and severe poor vision (≤4.5). The worst score from either eye was used for the study analysis.

The standard logarithmic visual acuity chart is only widely used in China. In order to compare with other research results, we listed the conversion formula of visual acuity score below [31]:L = 5 + logd/D
where L = the visual acuity score of the standard logarithmic visual acuity chart, d = the distance between the chart and the person’s eye and D = the furthest distance that a person with normal vision can see the optotype.

#### 2.2.2. Behavior

Data on reading habits, screen time, homework time, and sleep duration were included in the analysis. Students were asked to respond to multiple-choice questions. The question of reading habits was 1 item that surveyed the frequency of reading in bed in daily life. The response options were as follows: (1) never; (2) sometimes; (3) often, and (4) always. The screen time survey included two items. One focused on average daily screen time on weekdays, with the options of 0 min, 1–30 min, 31–60 min, 1–2 h, 2–3 h, and above 3 h. The other focused on average daily screen time on weekends, with the options of 0 min, 1–30 min, 31–60 min, 1–2 h, 2–3 h, 3–5 h, and above 5 h. The homework time spent on math homework during weekdays was surveyed in 1 item with options of no homework, 1–15 min, 16–30 min, 31–60 min, 1–2 h, and above 2 h. Sleep duration was calculated from 2 self-reported items. One item surveyed the daily wake-up time while the other item surveyed the daily go-to-bed time. According to the guidelines provided by the Chinese MOE for a healthy lifestyle in school-age students, primary school students should sleep for 10 h each day, while middle school students should sleep for 9 h each day. We recorded the sleep duration of students into the dichotomous variables of sleep sufficiency and sleep insufficiency.

### 2.3. Mathematics Achievement

Mathematics achievement came from a test developed by mathematics teachers and experts in mathematics education and educational measurement. The test involved three content strands including (1) numbers & algebra, (2) space & shape, and (3) statistics & uncertainty. The internal consistency of the booklets was 0.85 to 0.88. Each of the test booklets was composed of twelve multiple-choice items and six to nine construct response items. The Rasch model and concurrent calibration were used to link scores of different test booklets to an identical scale provided by Conquest 1.1 [32]. The item difficulty ranged from −2.84 to 3.56 logits. A new scale was generated ranging from 229 to 768 with a mean of 500 and a standard deviation of 100.

### 2.4. Statistical Analyses

The descriptive statistics were calculated using SPSS 21.0 for Windows (SPSS Inc., Chicago, IL, USA) and presented as the means and standard deviations or percentages (%) in the forms of tables and charts to show the prevalence and relations between poor vision and related behaviors. Adjusted standardized residuals for post-doc testing of chi-square were also shown.

Furthermore, a linear regression was conducted to investigate the reciprocal associations among the variables, which used a smaller sample compared to the descriptive analysis due to the rotation design. CNAEQ employed a rotation design [33] to interchange sample scale and time demands by reducing the individual questionnaire response time. The questionnaire included three Forms (A, B, C), and one was randomly assigned to each student. Thus, about 10 per school for each form. Details could be seen in Wu et al. [27]. Therefore, we only included students assigned to Form A into the regression model because only Form A included all the variables needed in the study. For accuracy reasons, we only included the sample from Grade 8 and excluded instances with any missing information. The final sample size included in the regression model was 22,402.

The sample information and descriptive statistics of the variances included in the current model were presented as means and standard deviations for scale variables, including academic performance, SES, and age, while the nominal and ordinal variables were presented as percentages (%), including sex, age, and school location, homework hours, reading in bed, and screen time during weekdays and weekends. Linear regression was conducted to investigate the reciprocal associations among the variables.

## 3. Results

Vision screening results showed that poor vision in 37.1% of the student in Grade 4, and 66.2% of those in Grade 8. With the increase of age, the prevalence of poor vision tended to increase significantly. The rates of poor vision at different levels also varied among students of different grades. The proportion of moderate poor vision was highest among the fourth graders, while the highest proportion of poor vision was severe poor vision among the eighth graders. Girls were more susceptible than boys with a higher prevalence and a higher proportion of severe poor vision. Details can be seen in Table 1 and Table 2.

The analysis between poor vision and vision-related risk behavior characteristics that included reading in bed, sleep time and average screen time during weekdays and weekends (see Table 1 and Table 2) suggested that students who reported a more frequent reading in bed behavior tended to have poor vision, especially in Grade 8. In 2017, China’s Ministry of Education issued the Standards for the Management of Compulsory Education Schools, which set requirements for sleep time for primary and secondary school students [34]. Students who failed to foster sleeping time standards set by the government tended to have poor vision, especially among the eighth graders. In the analysis of the effect of screen time on poor vision, we found that the students who had less screen time during weekends tended to have better vision. The post-doc testing results also confirmed and provided more detailed information.

In the analysis on the relationship between homework hours and mathematics achievement, an inverted U curve-like association could be observed between homework hours and mathematics achievements. There was a point that divides the trend of the association between vision and mathematics achievement. students of Grade 8 who spent 31–60 min on mathematics performed best in the mathematics assessment but had the lowest percentage of normal vision. There was a similar trend among the students of Grade 4, but the critical point was 15–30 min. Details can be seen in Figure 2 and Figure 3.

Before conducting the linear regression, we analyzed the characteristics of sample and the correlation between variables. The analysis results were shown in Table 3 and Table 4.

After controlling SES, sex, age, and school location as covariates, the linear regression estimated the associations between visual acuity, academic performance, homework hours, reading in bed, and screen time during weekdays and weekends. Due to the high correlation (0.700) between parents’ education status and SES, parents’ education status was not included in the regression as in other research. According to the final analysis results in Table 5, we can see that the students with poor mathematics achievement have poorer eyesight, and the longer homework time, the worse their vision will be. The more frequently the student read while lying in bed, the worse the vision will be. The more average screen time on weekends, the worse the vision was.

## 4. Discussion

In recent times, poor vision has been one of the most prevalent non-communicable diseases in terms of number, and it progresses into severe poor vision in school-age students in China. The present study provided an overall description of the prevalence of poor vision in Chinese Grade 4 and Grade 8 students and analyzed the relevant risk behaviors and mathematics achievements’ consequences on visual acuity based on the sample of Grade 8. Our findings substantiated previous reports that age was an important factor in poor vision progression and that females were more vulnerable than their male peers. The present study built on previous work by investigating risk behaviors and supported the proposal that prolonged academic-related tasks and bad reading habits are positively associated with poor vision [35]. The analysis of the consequences of vision in the current study suggested a negative association between vision and mathematics achievements.

Consistent with a previous demographic study, a dramatic increase in poor vision prevalence between Grade 4 and Grade 8 students indicated a tendency for poor vision being associated with increasing age in youth [36]. Previous studies have suggested that the prevalence of myopia exceeds 60% among 12-year-olds in China after elementary school, reaches nearly 80% at 16 years old after junior high school, and surpasses 90% in university students [37,38]. This tendency with age showed a continuous progression to severe poor vision at the older ages. Besides, girls had worse vision than boys. This finding was consistent with screenings and relevant surveillance in adolescents [39], although, to date, no appropriate explanation exists for this gender difference.

Previous studies have observed that students with poor vision were more likely to show poor academic performance in mathematics and English [20]. This association has been attributed to impaired visual acuity, which may reduce the capacity for children to perform optimally on visually demanding tasks, including distance and near changes, near tasks, and computer-based tasks in the modern classroom [11]. In the present study, there was a positive association between poor vision and academic performance, which revealed that students with poor vision were more likely to have better mathematics performance than their peers with normal vision. This result was consistent with previous evidence from China and other Southeast Asian countries. The inconsistent results from Southeast Asian countries and Peru could be attributed to the difference between the two countries on the learning process and learning content. Also, the inconsistent results between the current study and that of Uysal and Aki [20] are caused by the natural differences between writing and mathematics.

Previous studies have suggested homework time as a measure for learning efforts and obtained positive associations between homework time, poor vision, and academic performance. In the current analysis, average math homework time during weekdays, mathematics achievement, and poor vision were slightly but significantly associated. The descriptive statistic results suggested that students of Grade 8 who spent 31–60 min on mathematics performed best in the mathematics assessment but had the lowest percentage of normal vision. Spending 31–60 min daily for mathematics homework could be recognized as the critical point that divides the trend of the association between vision and mathematics achievement. Students who spent less than 30 min on mathematics homework had an inverse association in that the more time the students spent on homework, the better the mathematics achievement and the worse the vision impairment was. In contrast, students who spent more than one hour on mathematics homework showed another inverse association in that the more time students spent on homework, the worse the mathematics achievement and the less the vision impairment was. Similar trends were observed in Grade 4 students, but the critical point was 15–30 min. There was not a simple linear association between homework time and academic achievement, it was unreasonable to predict academic achievement with homework time. We proposed these non-linear associations caused the low coefficients. Previous studies suggested a complex correlation between homework and academic achievement [40]. Further, we proposed this non-linear association of students’ variation in achievement based on previous analyses of the association between visual acuity, IQ, and academic achievement [41]. This relationship indicated higher odds of poor vision in students with high education and academic achievement than in students with higher IQs and assumed a possible learning strategy of students with average IQs that they spent more time on learning and practice. Unfortunately, it is a general phenomenon that students in developing countries have sacrificed their vision for prolonged learning and close-up reading and writing for better academic achievements [42]. It has been believed that academic achievement was efficient support for youth to gain success in potential competition leading to a higher social-economic level in the future. In these cases, parents, teachers, and students themselves should focus on the amount of work needed for learning instead of attending to protecting the students’ healthy vision. Another explanation about this non-linear association is the shortage of our present variable. As an education assessment that specific for a certain discipline, the CNAEQ-MA only assessed the average homework time of mathematics instead of the average total homework time. Therefore, the estimation of average homework time in the current regression is not accurate enough to estimate students’ total academic burden. Considering the homework time allocation of middle school students, students who spend less time on mathematics could be the students who spent less time on total academic tasks or be the students who spent more time on total academic tasks than their peers.

Factor analysis results have revealed slight coefficients but significant associations between vision-related risk behaviors and poor vision. Previous studies have demonstrated inverse associations between sleep time, screen time, and poor vision. The descriptive results were in line with these demonstrations in students of Grade 8, but the regression coefficients were lower than we expected. According to the schedule differences between weekdays and weekends, we varied the responses of time ranges to investigate the influence of accumulated screen time to understand an accurate association between screen time and poor vision. There was a decrease in daily screen time in Grade 8 students compared to Grade 4 students since the middle school students spent more time on homework and extracurricular learning, which also consequently led to a decrease in sleep time. Screen time is recognized worldwide as a vision-impairing factor. Students who spent more time looking at a screen were at a higher risk of poor vision. Inconsistent with previous studies, the regression excluded the sleeping variable for model fitness reason. The descriptive results of students’ sleep duration were based on the criteria set by government guidance. The criterion for Grade 4 students was 1 h more sleep than for Grade 8 students. The descriptive results suggested that Grade 8 students with sufficient sleep had a higher percentage of normal vision than those who failed to follow the recommended sleep guidelines, which was not observed in Grade 4 students. In that case, the effect of sufficient sleep on poor vision impairment remains to be discussed. We hypotheses this low coefficient and insignificant estimation for healthy lifestyle-related factors are protectable factors, which may not gain equal predictive effect with risk factors.

Although this study has some limitations, the current results illustrate the influence of academic-related risk behaviors on students’ vision. We hope that the results of this study will further promote the joint efforts of government departments, schools, parents and healthcare professionals to ensure the healthy growth of children. It is essential to realize that the behavior of extending students’ homework time may cause the increasing degree of vision impairment in the future. It should be ensured that children get enough sleep every day and maintain healthy lifestyle and academic-related habits, ensuring that children can access to a healthy and happy future.

### Limitations

The present study provided the overall prevalence of poor vision among school-aged students in China and its association with vision-related risk behaviors and mathematics achievements. The study involved a stratified sample on a large scale with national representation. Though there are significant associations between poor vision, reading in bed, homework hours, screen time, and mathematics achievements, this study was not sufficient to determine the cause and effect relationships. The reason might be the high impact of inheritance on vision. In the present study, we investigated risk behaviors through academic-related behaviors. However, we included only the mathematics-relevant homework without accounting for the total time for reading and learning and the reading distance. Though it did not cover all aspects of visually demanding tasks, there was enough data to associate these tasks with the poor vision prevalence and the progression to severe poor vision.

## 5. Conclusions

The prevalence of poor vision in elementary and middle school students was at a high proportion and progressed to severe poor vision with increasing age. Poor vision was positively associated with mathematics achievements but harmful to students’ potential educational opportunities. This study highlighted the importance of vision-related risk behaviors, including avoiding reading in bed, prolonged homework, and screen time. It also demonstrated the urgent need for integrated vision screening into an in-school physical assessment at least once per year.

## Figures and Tables

**Figure 1 ijerph-17-08561-f001:**
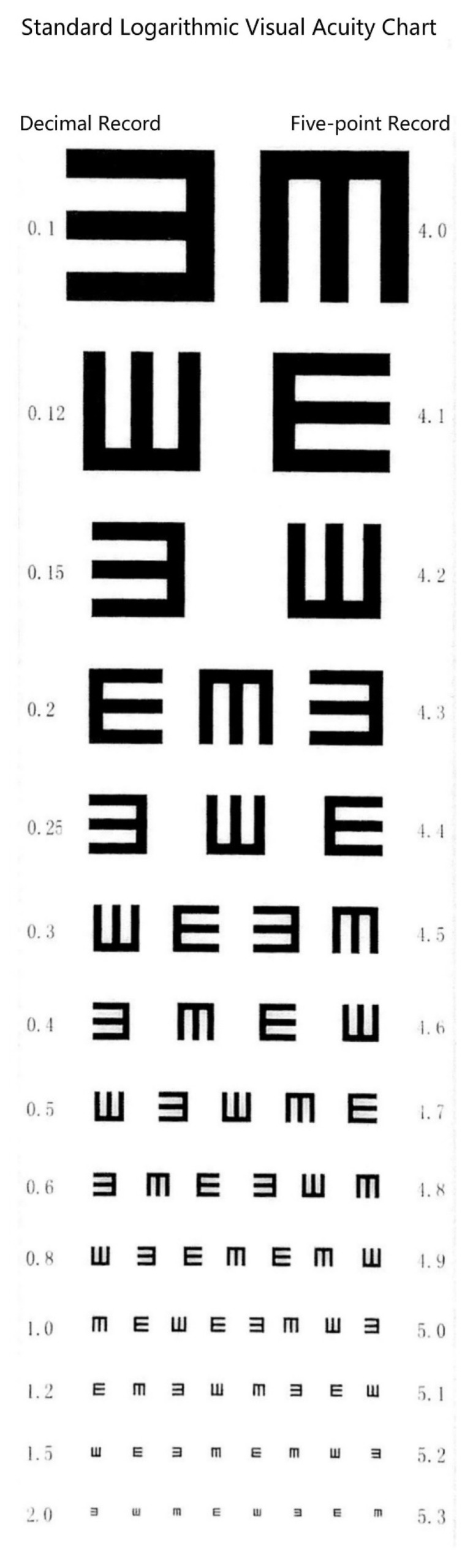
The Standard Logarithmic Visual Acuity Chart.

**Figure 2 ijerph-17-08561-f002:**
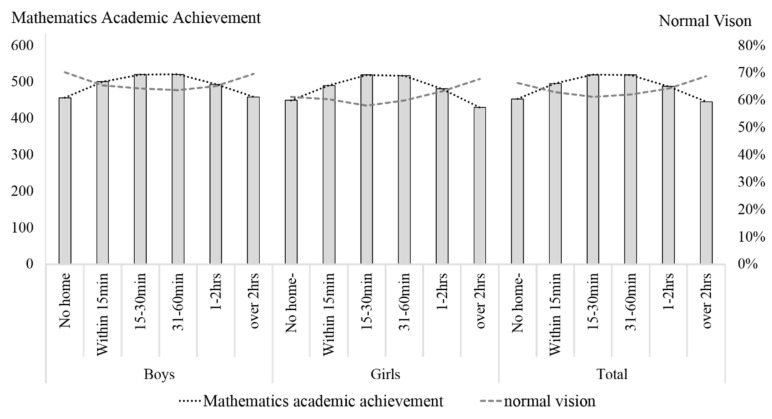
Mathematics Achievement and Prevalence of Normal Vision Based on Hours of Homework in Grade 4.

**Figure 3 ijerph-17-08561-f003:**
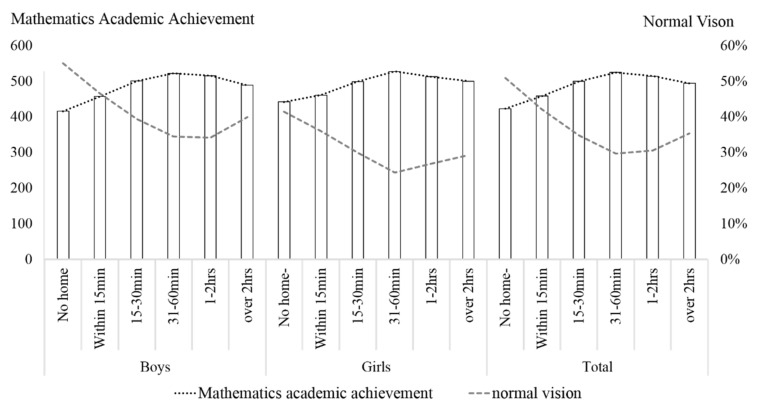
Mathematics Achievement and Prevalence of Normal Vision Based on Hours of Homework in Grade 8.

**Table 1 ijerph-17-08561-t001:** Poor Vision and Risk Behavior Characteristics of Grade 4 Students by Sex.

Variables	Boys				Girls				Total			
	1 ^a^	2	3	4	1	2	3	4	1	2	3	4
Poor vision	65.10%	9.10%	16.70%	9.20%	60.50%	10.00%	18.90%	10.60%	62.90%	9.50%	17.70%	9.90%
Reading in bed												
never	66.10%	9.20%	16.20%	8.50%	62.20%	10.20%	18.10%	9.60%	64.10%	9.70%	17.20%	9.10%
	(5.1) ^b^	(−0.6)	(−2.9)	(−4.1)	−6.8	−0.1	(−3.5)	(−6.4)	−7.8	(−0.1)	(−4.2)	(−7.1)
sometimes	63.50%	9.40%	17.30%	9.80%	58.70%	10.30%	19.30%	11.60%	61.30%	9.80%	18.30%	10.60%
	(−4.7)	−0.8	−2.4	−3.8	(−5.7)	−0.7	−2.4	−5.4	(−7)	−0.9	−3.2	−6.3
usually	64.40%	9.00%	17.20%	9.40%	57.90%	7.80%	22.00%	12.40%	61.80%	8.50%	19.10%	10.60%
	(−0.3)	(−0.2)	−0.3	−0.3	(−1.6)	(−2.3)	−2.4	−1.7	(−0.9)	(−1.8)	−1.7	−1.1
always	61.80%	8.60%	19.10%	10.50%	51.90%	9.30%	23.70%	15.10%	58.40%	8.80%	20.60%	12.10%
	(−1.5)	(−0.6)	−1.4	−1.1	(−3)	(−0.5)	−2.1	−2.6	(−2.6)	(−0.9)	−2.2	−2.2
Sleep time												
less than 7 h	66.70%	10.20%	15.60%	7.50%	73.60%	7.90%	13.10%	5.40%	69.10%	9.40%	14.70%	6.80%
	−0.9	−0.7	(−0.8)	(−1.2)	−4.1	(−1.2)	(−2.2)	(−2.5)	−3.5	(−0.3)	(−2.1)	(−2.6)
7~8 h (inc 7 h)	68.60%	8.80%	15.60%	7.10%	63.30%	9.70%	17.80%	9.20%	66.50%	9.10%	16.50%	7.90%
	−3.1	(−0.6)	(−1.4)	(−2.8)	−1.9	(−0.5)	(−0.8)	(−1.4)	−3.9	(−0.9)	(−1.7)	(−3.2)
8~9 h (inc 8 h)	67.40%	9.40%	15.90%	7.40%	63.00%	9.90%	17.80%	9.30%	65.40%	9.60%	16.80%	8.30%
	−5.4	−0.4	(−2.6)	(−6)	−4.6	(−0.7)	(−2.3)	(−3.6)	−7.3	(−0.3)	(−3.6)	(−6.9)
9~10 h (inc 9 h)	63.50%	9.30%	17.40%	9.80%	59.10%	10.40%	19.40%	11.00%	61.30%	9.90%	18.40%	10.40%
	(−3.6)	−0.2	−2.2	−3	(−4.3)	−1.7	−2.3	−2.2	(−5.9)	−1.5	−3.4	−3.8
more than 10 h (inc 10 h)	63.70%	9.10%	17.10%	10.00%	60.10%	9.90%	18.90%	11.00%	62.00%	9.50%	18.00%	10.50%
	(−2.1)	(−0.5)	−0.6	−3.3	(−0.5)	(−0.9)	−0.1	−1.5	(−1.8)	(−1)	−0.5	−3.4
Standard sleep time												
less than 10 h	64.90%	9.30%	16.90%	8.90%	60.40%	10.20%	18.90%	10.50%	62.70%	9.80%	17.90%	9.70%
	−2.1	−0.5	(−0.6)	(−3.3)	−0.5	−0.9	(−0.1)	(−1.5)	−1.8	−1	(−0.5)	(−3.4)
10 h or more	63.70%	9.10%	17.10%	10.00%	60.10%	9.90%	18.90%	11.00%	62.00%	9.50%	18.00%	10.50%
	(−2.1)	(−0.5)	−0.6	−3.3	(−0.5)	(−0.9)	−0.1	−1.5	(−1.8)	(−1)	−0.5	−3.4
Average screen time during weekdays												
0 min	63.30%	9.20%	17.40%	10.10%	60.20%	9.90%	19.20%	10.70%	61.70%	9.50%	18.30%	10.40%
	(−4.1)	(−0.2)	−2	−4.4	(−0.6)	(−1.4)	−1.5	−0.4	(−3.9)	(−1)	−2.8	−3.7
1–30 min	65.10%	9.10%	17.00%	8.70%	60.60%	10.30%	18.50%	10.70%	62.80%	9.70%	17.70%	9.70%
	−1.2	(−0.5)	−0.6	(−2.2)	−0.6	−0.6	(−1.4)	−0.3	−0.7	−0.3	(−0.3)	(−1.1)
30 min–1 h	63.20%	9.90%	17.10%	9.80%	58.00%	10.50%	19.80%	11.70%	61.10%	10.20%	18.20%	10.60%
	(−2.4)	−1.8	−0.4	−1.7	(−3)	−0.6	−1.6	−2.1	(−3.1)	−1.5	−0.9	−2.3
1–2 h	68.80%	8.70%	15.20%	7.30%	63.70%	11.10%	17.40%	7.80%	66.80%	9.60%	16.10%	7.50%
	−4.3	(−1)	(−2.2)	(−3.2)	−2.7	−1.2	(−1.4)	(−3.7)	−5.6	(−0.2)	(−2.9)	(−5.1)
2–3 h	67.80%	9.70%	14.00%	8.50%	59.70%	10.20%	17.60%	12.40%	65.10%	9.90%	15.20%	9.80%
	−2.3	−0.6	(−2.8)	(−0.8)	(−0.4)	−0.1	(−0.8)	−1.5	−2.2	−0.3	(−2.9)	(−0.1)
over 3 h	68.20%	9.00%	15.60%	7.30%	65.40%	9.20%	18.20%	7.20%	67.40%	9.00%	16.30%	7.20%
	−3	(−0.4)	(−1.4)	(−2.8)	−2.5	(−0.7)	(−0.4)	(−2.8)	−4.8	(−1.1)	(−1.9)	(−4.3)
Average screen time during weekends												
0 min	69.50%	8.60%	13.60%	8.20%	66.50%	9.70%	15.70%	8.20%	67.80%	9.20%	14.80%	8.20%
	−7.6	(−1.6)	(−6.5)	(−2.5)	−10.9	(−1.4)	(−7)	(−6.9)	−12.3	(−1.8)	(−9.1)	(−6.5)
1–30 min	65.40%	9.30%	16.10%	9.20%	59.60%	10.00%	19.50%	11.00%	62.30%	9.70%	17.90%	10.20%
	−1.9	(−0.1)	(−2.4)	−0.2	(−2.5)	(−0.7)	−2.4	−1.7	(−1.6)	(−0.2)	−0.7	−1.9
30 min–1 h	61.60%	9.40%	19.40%	9.60%	57.70%	10.30%	19.90%	12.10%	59.80%	9.80%	19.60%	10.80%
	(−6.5)	−0.5	−6.7	−1.4	(−5.1)	−0.4	−2.6	−4.4	(−8)	−0.6	−6.5	−4
1–2 h	62.70%	9.00%	17.90%	10.40%	57.60%	10.80%	20.20%	11.30%	60.60%	9.80%	18.90%	10.80%
	(−3.1)	(−0.8)	−2.2	−3.1	(−3.7)	−1.4	−2.3	−1.5	(−4.2)	−0.2	−2.8	−3
2–3 h	64.00%	9.60%	17.20%	9.20%	60.30%	10.60%	19.00%	10.10%	62.60%	10.00%	17.90%	9.50%
	(−0.9)	−0.6	−0.7	0	(−0.1)	−0.8	−0.2	(−0.8)	(−0.1)	−0.7	−0.3	(−0.8)
3–5 h	65.50%	9.80%	15.90%	8.80%	60.60%	10.60%	18.30%	10.50%	63.90%	10.10%	16.60%	9.40%
	−0.8	−0.9	(−1.3)	(−0.6)	−0.1	−0.5	(−0.5)	(−0.1)	−1.6	−0.7	(−1.8)	(−1)
over 5 h	66.10%	9.90%	16.70%	7.30%	62.60%	10.30%	17.30%	9.80%	65.20%	10.00%	16.90%	8.00%
	−1.3	−1	(−0.2)	(−3.1)	−1.2	−0.1	(−1.1)	(−0.7)	−2.8	−0.6	(−1.3)	(−3.5)

^a^ 1 = normal vision, 2 = mild poor vision, 3 = moderate poor vision, 4 = severe poor vision; ^b^ Adjusted standardized residuals for post-doc testing of chi-square, considered as significant when absolute values were above 3.

**Table 2 ijerph-17-08561-t002:** Poor Vision and Risk Behavior Characteristics of Grade 8 Students by Sex.

Variables	Boys				Girls				Total			
	1 ^a^	2	3	4	1	2	3	4	1	2	3	4
Poor vision	38.50%	6.90%	22.30%	32.30%	28.50%	7.00%	24.40%	40.20%	33.80%	6.90%	23.20%	36.00%
Reading in bed												
never	46.10%	7.20%	20.40%	26.20%	31.70%	7.40%	24.10%	36.90%	40.60%	7.30%	21.80%	30.30%
	(15) ^b^	−1.6	(−4.6)	(−12.4)	−5	−1.3	(−0.8)	(−4.6)	−17.2	−2	(−4.6)	(−14)
sometimes	36.90%	6.60%	23.00%	33.50%	28.70%	6.90%	24.70%	39.70%	32.70%	6.80%	23.90%	36.60%
	(−6.8)	(−1.5)	−2.8	−5.4	−0.6	0	−0.7	(−1.1)	(−6.6)	(−1.1)	−3	−4.5
usually	28.50%	7.20%	24.00%	40.30%	22.90%	6.40%	25.50%	45.20%	25.70%	6.80%	24.80%	42.80%
	(−10.1)	−0.8	−1.9	−8.4	(−6.2)	(−1.1)	−1.1	−5.3	(−11.9)	(−0.2)	−2.2	−9.9
always	28.30%	5.00%	24.90%	41.80%	21.30%	5.20%	19.00%	54.50%	25.00%	5.10%	22.20%	47.80%
	(−4.4)	(−1.5)	−1.2	−4.3	(−3.2)	(−1.4)	(−2.5)	−5.8	(−5.3)	(−2.1)	(−0.9)	−7.1
Sleep time												
less than 7 h	35.50%	6.90%	22.30%	35.30%	25.40%	6.10%	23.90%	44.60%	29.90%	6.50%	23.20%	40.40%
	(−2.6)	−0.3	(−0.1)	−2.7	(−3.3)	(−1.5)	(−0.8)	−4.6	(−5.2)	(−0.9)	(−0.4)	−6
7~8 h (inc 7 h)	36.70%	7.30%	22.10%	34.00%	27.10%	7.40%	24.80%	40.60%	31.70%	7.30%	23.50%	37.40%
	(−3.4)	−1.7	(−0.8)	−3.3	(−3)	−2.2	−0.5	−1.2	(−5.8)	−2.7	−0.1	−4.2
8~9 h (inc 8 h)	38.40%	6.60%	22.90%	32.10%	29.30%	6.80%	24.20%	39.70%	34.20%	6.70%	23.50%	35.60%
	(−0.1)	(−1)	−1.6	(−0.8)	−2.3	(−0.7)	(−1.1)	(−0.9)	−2	(−1.2)	−0.2	(−1.6)
9~10 h (inc 9 h)	41.90%	6.50%	21.60%	30.00%	30.60%	6.20%	26.10%	37.10%	37.40%	6.40%	23.40%	32.80%
	−4.8	(−0.9)	(−1.3)	(−3.4)	−2.6	(−1.4)	−1.9	(−3.3)	−6.8	(−1.6)	(−0.1)	(−5.8)
more than 10 h (inc 10 h)	44.10%	6.70%	23.40%	25.80%	36.30%	9.20%	23.10%	31.40%	41.30%	7.60%	23.30%	27.90%
	−2.9	(−0.1)	−0.6	(−3.5)	−3.3	−1.7	(−0.6)	(−3.4)	−5.1	−1	(−0.2)	(−5.4)
Standard sleep time												
less than 9 h	37.50%	6.90%	22.60%	33.00%	28.00%	6.90%	24.40%	40.60%	32.80%	6.90%	23.50%	36.80%
	(−5.8)	−0.9	−1	−4.6	(−3.7)	−0.7	(−1.6)	−4.4	(−8.4)	−1.2	−0.1	−7.6
9 h or more	42.20%	6.50%	21.90%	29.40%	31.30%	6.60%	25.70%	36.40%	37.90%	6.50%	23.40%	32.20%
	−5.8	(−0.9)	(−1)	(−4.6)	−3.7	(−0.7)	−1.6	(−4.4)	−8.4	(−1.2)	(−0.1)	(−7.6)
Average screen time during weekdays												
0 min	36.30%	7.10%	22.90%	33.70%	28.50%	6.90%	25.40%	39.20%	32.30%	7.00%	24.20%	36.50%
	(−4.1)	−0.9	−1.1	−2.8	(−0.1)	−0.1	−1.7	(−1.5)	(−4.1)	−0.8	−2.3	−1.7
1–30 min	37.40%	6.80%	21.90%	34.00%	27.20%	7.10%	24.30%	41.40%	32.10%	6.90%	23.20%	37.80%
	(−2.4)	(−0.1)	(−1.1)	−3.6	(−3.1)	−0.7	(−0.6)	−3.1	(−5.2)	−0.4	(−0.9)	−5.7
30 min–1 h	39.40%	6.40%	23.10%	31.20%	29.50%	6.90%	23.70%	39.90%	34.80%	6.60%	23.40%	35.30%
	−1.1	(−1.3)	−1.2	(−1.6)	−1.5	(−0.1)	(−1.4)	(−0.1)	−2	(−1)	(−0.2)	(−1.4)
1–2 h	40.30%	7.00%	21.30%	31.40%	29.20%	6.90%	25.00%	38.90%	35.70%	6.90%	22.90%	34.50%
	−2	−0.3	(−1.4)	(−1)	−0.7	0	−0.5	(−1)	−2.9	−0.2	(−1)	(−2.2)
2–3 h	39.90%	6.50%	23.80%	29.90%	29.10%	5.80%	25.70%	39.40%	35.70%	6.20%	24.50%	33.60%
	−1.1	(−0.6)	−1.4	(−2.1)	−0.4	(−1.4)	−0.8	(−0.4)	−2.1	(−1.3)	−1.4	(−2.6)
over 3 h	43.40%	7.10%	21.40%	28.10%	32.20%	6.70%	23.20%	38.00%	39.60%	7.00%	22.00%	31.50%
	−5	−0.5	(−1.1)	(−4.5)	−2.9	(−0.3)	(−1.2)	(−1.5)	−7.6	−0.3	(−2.1)	(−5.8)
Average screen time during weekends												
0 min	47.50%	8.90%	19.80%	23.90%	37.70%	7.80%	18.80%	35.70%	42.60%	8.40%	19.30%	29.70%
	−3.9	−1.7	(−1.3)	(−3.9)	−4.4	−0.7	(−2.9)	(−1.8)	−5.7	−1.8	(−3)	(−3.9)
1–30 min	46.20%	6.40%	20.90%	26.60%	33.20%	7.70%	25.50%	33.60%	38.50%	7.20%	23.60%	30.80%
	−7	(−0.8)	(−1.6)	(−5.4)	−5.7	−1.9	−1.2	(−7.2)	−7	−0.9	−0.3	(−7.7)
30 min–1 h	40.20%	6.80%	21.50%	31.40%	30.30%	6.80%	23.40%	39.50%	34.60%	6.80%	22.60%	36.00%
	−2.1	0	(−1.3)	(−1.1)	−2.7	(−0.4)	(−1.9)	(−0.6)	−1.6	(−0.2)	(−1.8)	−0.2
1–2 h	37.00%	6.10%	22.80%	34.20%	25.80%	6.90%	24.40%	42.90%	31.40%	6.50%	23.60%	38.50%
	(−2.4)	(−2.2)	−0.7	−3.1	(−4.5)	(−0.1)	(−0.3)	−4.5	(−5.3)	(−1.7)	−0.4	−5.7
2–3 h	36.60%	6.60%	22.90%	34.00%	25.60%	6.90%	24.20%	43.20%	31.50%	6.70%	23.50%	38.30%
	(−3.3)	(−0.8)	−1	−3	(−4.8)	−0.1	(−0.6)	−4.9	(−5.4)	(−0.5)	−0.2	−5.4
3–5 h	37.20%	7.30%	22.20%	33.30%	27.30%	6.40%	27.00%	39.30%	33.20%	6.90%	24.20%	35.80%
	(−2)	−1.4	(−0.3)	−1.6	(−1.6)	(−1.3)	−3.3	(−0.7)	(−1.3)	−0.2	−1.6	(−0.3)
over 5 h	38.40%	7.40%	23.10%	31.10%	30.50%	6.70%	24.60%	38.20%	35.50%	7.10%	23.70%	33.60%
	(−0.3)	−1.6	−1.3	(−1.8)	−2.2	(−0.4)	−0.1	(−1.9)	−3.3	−1	−0.5	(−4.3)

^a^ 1 = normal vision, 2 = mild poor vision, 3 = moderate poor vision, 4 = severe poor vision; ^b^ Adjusted standardized residuals for post-doc testing of chi-square, considered as significant when absolute values were above 3.

**Table 3 ijerph-17-08561-t003:** Characteristics of the sample included in the analysis.

Variables	Mean/Percentage	SD
Vision (mean and SD)	4.70	(0.37)
Maths (mean and SD)	503.97	(96.52)
Average maths homework time during weekdays (%)		
1. No homework	2.5	
2. Within 15 min	11.6	
3. 15–30 min	36.4	
4. 30–60 min	30.6	
5. 1–2 h	14.3	
6. Over 2 h	4.6	
Average screen time during weekdays (%)		
1. 0 min	25.6	
2. 1–30 min	31.4	
3. 30–60 min	18.2	
4. 1–2 h	10.9	
5. 2–3 h	6.1	
6. Over 3 h	7.8	
Average screen time during weekends (%)		
1. 0 min	2.0	
2. 1–30 min	9.9	
3. 30–60 min	16.7	
4. 1–2 h	19.7	
5. 2–3 h	21.0	
6. 3–5 h	15.1	
7. Over 5 h	15.5	
Reading in bed (%)		
1. Never	23.9	
2. Sometimes	65.1	
3. Usually	9.3	
4. Always	1.7	
SES (mean and SD)	0.06	(0.98)
Gender (female%)	49.0	
Age (mean and SD)	13.94	(0.64)
Location		
1. Urban	33.1	
2. County	22.0	
3. Country	44.9	

**Table 4 ijerph-17-08561-t004:** Correlations between variables.

	1	2	3	4	5	6	7	8	9
Vision									
Maths	−0.291								
Average maths homework time during weekdays	−0.095	0.182							
Average screen time during weekdays	0.047	−0.149	−0.016						
Average screen time during weekends	−0.023	0.014	0.027	0.474					
Reading in bed	−0.095	0.123	0.053	0.083	0.193				
SES	−0.229	0.321	0.139	0.075	0.118	0.070			
Gender	−0.116	0.032	0.023	0.117	−0.133	0.072	−0.018		
Age	0.094	−0.195	−0.041	0.039	−0.017	−0.044	−0.205	−0.066	
Location	0.197	−0.243	−0.113	−0.078	−0.040	−0.056	−0.429	0.022	0.106

All correlations are significant at *p* < 0.01.

**Table 5 ijerph-17-08561-t005:** OLS model standardized coefficients for vision.

	Standardized Coefficients	Unstandardized Coefficients	t	η^2^
Maths	−0.210 *	−0.001	−30.32	0.046
Average maths homework time during weekdays	−0.024 *	−0.008	−3.78	0.001
Average screen time during weekdays	0.034 *	0.008	4.66	0.001
Average screen time during weekends	−0.024 *	−0.006	−3.32	0.001
Reading in bed	−0.045 *	−0.027	−7.00	0.003
SES	−0.115 *	−0.043	−15.78	0.106
Gender	−0.108 *	−0.080	−17.06	0.012
Age	0.008	0.005	1.25	0.001
Location	0.094 *	0.040	13.48	0.007
Observation	22402			
R-squared	0.128			
F-test	365.699			

* *p* < 0.01.

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
