# Peer review of "Associations between Poor Vision, Vision-Related Behaviors and Mathematics Achievement in Chinese Students from the CNAEQ-PEH 2015"

_ijerph, 2020, doi:10.3390/ijerph17228561_

Round 1
Reviewer 1 Report
I appreciate the effort of the authors in the review of the manuscript
This manuscript is a resubmission of an earlier submission. The following is a list of the peer review reports and author responses from that submission.
Round 1
Reviewer 1 Report
Title. Association between poor vision, vision-related behaviors and mathematics achievement in Chinese students from the
The present work has relevance for the educational practices. Also, the study is clear and well written, and includes an important sample size. I have some minor considerations.
Abstract.
Avoid redundancies as “was conducted” in the abstract
The conclusions reached in the abstract are very surprising. Perhaps the authors want to soften these statements and explain them later in the Discussion. Also, a brief explanation of this result in the abstract could help to understand the conclusions reached.
“Poor vision is prevalent in school-aged students and irreversibly impairs overall health in children and youth” maybe the authors want to specify that poor vision is prevalent in school-aged students in some countries or specify in which countries or in which percentages.
Introduction
Lines 45-48. Authors indicate the hight prevalence of the poor vision. It could be appropriate to include the specific percentages reported by the previous studies.
Line 57. Authors point out that “…higher prevalence of vision impairment in girl students is attributed…” However, nothing was mentioned before in relation with the higher levels of poor vision in girls. It could be relevant to include first a paragraph which include specifically the relevant aspect regarding the prevalence of poor vision.
Line 60. “researchers proposed”. It would be convenient to include the cite to these reseachers. Also, line 61 “Observation results and some follow-up evidence supported this hypothesis” what results?
Line 78. “latest results from education follow-up”. In my view, it would be appropriated to cite what latest results.
Line 107. There is a typo “However. Based”
In my view, is possible to summarize the information in relation with the China National Assessment of Education Quality
The aim of the present work could be more detailed. Also, it could be relevant to include specifically, what does this study contribute to the literature?
Material and method
Line 223. “It included approximately” What do the authors want to say with “approximately”?
The description of the participants should be incorporate the media and standard deviation of age, and the number of girls and boys. Furthermore, it could be interesting to know if students with special education need or learning difficulties, were included in the sample or not
The instruments used are correctly described.
Results and discussion
Tables are difficult to understand. For example, in table 1, the 68.6% of the boys with poor vision (1) sleep less than 7 hours; the 68.6% sleep between 7-8 hours; and the 67.4% sleep between 8-9 hours. It does not have sense for me.
There is a typo in table 1 and table 2, second column (1ª instead of 1).
I suggest reviewing the presentation of statistics. Also, I suggest modifying the table 5 including the standardized and non-standardized β coefficients; the student t-test and the Cohen d for the effect size. Were all the variables included in the same model? Did the authors consider introducing the variables in two models?
In my opinion, it would be appropriate to include the tables separately with a description of the information instead of all the tables together and followed. For example, the authors can provide the matrix correlation and describe it and after the description the regression analysis and the explanation of the table.
The discussion could include the implications for the educational practice of these results. How this information could help to prevent poor vision? What practices should incorporate teachers and families regarding the sleep time or homework time?
Reviewer 2 Report
Review: Association between poor vision, vision-related behaviors and mathematics achievement in Chinese students from the CNAEQ-PEH 2015
The authors present a paper looking at behavioural risk factors for vision impairment (reading in bed, sleep, screen time, homework time) as well as the association between vision impairment and mathematics outcomes. The strength of the study is its scale and the standardised assessment of mathematics achievement.
I do have some concerns with the manuscript; primarily the presentation of the results and the analysis. The introduction is also very long, and considering the results section is currently very brief, I think the balance of the manuscript is out.
With the results, perhaps a better structure would be participant demongraphics, risk factors for vision, and then role of vision on maths achievement (regression). With the risk factors section (Tables 1 and 2), all of the percentages are very difficult to interpret (and the p-values). Could the n’s be reported, and or flipping how the results are presented so that for children with vision impairment = 4, we can see the spread of use of screen time, for children with VI = 3, and so on…
I think the regression presented in Table 5 is also still looking at the risk factors for vision; rather than looking at the role of vision on mathematics achievement which was the second aim of the study. I think this regression needs to be re-done so that mathematics is the outcome variable.
Comments by section are below.
Abstract:
Line 12. Opening sentence doesn’t link very well to the next sentence (study aim). Could the first sentence be extended to discuss other outcomes as well that poor vision can impact, e.g. educational. It would also be nice to see an example of what ‘overall health’ is affected by poor vision.
Line 21. Can the cut-off used to quantify poor vision be included in the abstract as well.
Line 25. Can Academic achievements be changed to mathematics, as this was the only academic parameter assessed.
Line 27. Is compulsory needed?
Introduction.
It would be nice to see a definition of vision impairment somewhere in the first few sentences of the introduction; e.g. a reduction in visual acuity of 6/12 or more?? Throughout the review of other studies it would also be nice to report what VA cut-offs were used across studies to define vision impairment (and whether this was habitual or unaided).
I would like the authors to explore more key differences in what is likely to be causing vision impairment in the different studies in children; which may explain why some show an association with better academic achievement and others with poorer academic achievement. If the vision impairment is largely driven by uncorrected or undercorrected myopia it would make sense that this is associated with better academic outcomes because the near visual function is unaffected, if anything, advantaged because of the reduced accommodative demand. Whereas, if the vision impairment is due to a combination of hyperopia, astigmatism and myopia, then near vision may also be effected – hence resulting in poorer academic outcomes.
Line 116-131. All of these factors are really relating to myopia and myopia progression as the cause for vision impairment. In countries which have higher proportions of hyperopia and astigmatism in children (and less myopia), this would not be relevant. Please take care when discussing vision impairment more broadly, and that related specifically to a measurable reduction in distance acuity due to un/under corrected myopia.
The introduction is very long; I wonder whether some of the literature review is better off in the discussion? Or putting the key studies as a table.
Line 34. Poor vision is a prolonged and progressive… change this to Poor vision can be a prolonged and progressive…
Line 35. ‘can have’ harmful effects on students’ academic performance,
Line 36. Remove ‘even’.
Line 39. What is meant by birth season?
Line 58. According to the… this sentence doesn’t make sense to me. Are the authors suggested that because increased academic effort means increased environmental/behavioural risk factors?
Line 61. Observation results… can this sentence be explained more? What is meant by observation results.
Line 62. ‘scored better academically’ rather than ‘won better academic achievements’
Line 71. ‘students with myopia’ not ‘myopias’; and change as well ‘students with emmetropia or hyperopia’
Line 72. This sentence needs re-wording; all of the previous studies 15 – 16 mentioned are correlational, so it can not be surmised that it is due to duration spent on academic tasks. Indeed, some children with higher academic scores may be spending less time doing academic work than poorer achieving students.
Line 74. Students with high academic… do the authors have references to support this? Or is this in their opinion. If the latter, I think they need to make the comment that amount of time doing schoolwork is not linearly proportional to academic achievement. Some students are more naturally gifted, and don’t need to work as hard as others to score well, whereas some children work very hard, ineffectively.
Line 78. Change end of sentence to (and merge with following sentence) ‘suggest an inconsistent association between academic achievement and vision impairment’; with students with poor vision performing worse than their peers with healthy vision.
Lines 88 – 93. Are there references?
Lines 95. Ref 20 – this needs more explaining if to be included; age group, country; n;
Lines 97 – 103. What is the reference for this research? Also, can you critique their statements more? Are they based on a solid research study – or their own opinions.
Line 105. What is meant by sample scales? Demographics or characteristics, not both and not characters. What is meant by learning progress, and what about type of vision impairment?
Line 107. Remove However.
Line 107. Change to visual acuity
Line 109. Screen duration not ‘screening duration’
Line 114. Maybe ‘understand’ instead of ‘dig out’.
Line 117. Vision impairment is the term that has been used through the paper, not eye-impairment.
Line 118. Vision input-related habits (not eye) and Line 120.
Line 127. More critique of these studies would be beneficial. Daily consumption of eggs? Can the authors show the direction of the association and whether they feel the study design was strong?
Methods.
Lines 221 – 235 are very repetitive with the description of the CNAEQ in the Introduction.
Line 243. If the chart is logarithmic progression the line spacing won’t be equal across the lines. Can you also report the VA range of the charge in logMAR notation. Can the manufacturer of the VA chart also be reported.
Line 246. A line with arithmetical progression… I’m not sure what this means. Can this be explained more? Is there an image of the chart that can be included?
Line 250. Is 5m correct? Most charts are 3 or 6m.
Line 253. What was the termination rule? And were all letters pointed to on the lines near threshold?
Were children tested monocularly and binocularly?
Line 256. Can these levels and units be also shown as logMAR notation, or 6/6 equivalents.
Results.
Tables 1 and 2. What are the p-values relating to? Any post-hoc analysis? Wouldn’t it be of more value to compare the spread of sleep time in children by their vision impairment. Rather than compare the spread of vision impairment by sleep time category.
Line 324. Was this statistically significant? It also looks like lack sleep time and more screen time were less common in the vision impaired group?
Where are Figs 1 – 5?
Table 5. Wouldn’t the regression be to see what is contributing to variation in maths achievement (not vision)? It looks like vision has been put as the outcome variable, and not maths achievement.
Overall, I am not sure I am making the same conclusions that the authors are in Lines 324 – 338 based on the results presented in Tables 1 – 5. The Figures are also missing. There is a lot of data in Tables 1 and 2, and I wonder whether this can be simplified, as well as the key findings being drawn out (in the text).
Discussion.
Lines 356-358. Is this based on Table 5 of your results? I disagree with this statement based on the results you have presented. Further statistical analysis needs to be completed.
Line 376 – 376. As well as the differences in vision conditions resulting in vision impairment (myopia v other).
Line 389 – 394. I don’t agree with your suggestions here; perhaps children doing more maths homework are doing this because they are underachieving and find it more difficult to do maths, so it takes longer? I’m concerned about using homework time as a correlate of academic performance.
Conclusions.
Line 460. Authors conclude that reading in bed, prolonged homework and screen time are vision-related risk factors. Is this based on Table 5? And are all of the directions consistent – how come screentime weekdays coefficient is positive, but weekends it is negative? If it is based on Table 1, screen time looks like it reduces with vision impairment.